# The tracer nudging method for correcting and preventing uneven tracer distributions in geodynamical models

Paul James Tackley<sup>1</sup>

<sup>1</sup>Department of Earth and Planetary Sciences, ETH Zurich, Zurich, 8092, Switzerland

5 Correspondence to: Paul J. Tacklev (ptacklev@ethz.ch)

Abstract. Tracers/markers/particles are commonly used in geodynamical models to track composition and sometimes other quantities throughout the domain. A common problem is that over time, gaps in the tracer distribution can develop, often resulting in cells with no tracers as well as bunching of tracers. These arise when tracer advection does not perfectly respect the mass conservation equation, so here this equation is used to derive a correction method that perturbs or "nudges" the positions of tracers in such a way as to close gaps and eliminate bunching. Test results show that this tracer nudging method is highly effective. Starting from an extremely heterogeneous tracer distribution with large regions of the domain devoid of tracers, it can produce an even distribution in only a few nudge iterations. In a time-stepping situation with a nudge every time-step, the amplitudes of the nudges are small yet sufficient to prevent gaps and bunches, allowing a low-order tracer advection method to be used while maintaining a tracer distribution that is more even than that obtained using higher-order advection methods alone. The computational cost is small – slightly larger than that of a first-order tracer advection step alone - because the method simply requires solving a Poisson equation. If an accuracy threshold is applied, a nudge correction may be necessary in only a fraction of time steps, with tests indicating that it is fastest to use low order advection with more frequent nudges than high order advection with less frequent nudges.

## 1 Introduction

Tracers, alternatively named markers or particles, are commonly used in geodynamical models to track composition and sometimes temperature and other quantities, typically in the framework of a so-called "marker-and-cell" or "particle-in-cell" method, in which velocity and pressure are calculated on a fixed Eulerian grid while various other quantities are advected on Lagrangian tracers/markers/particles (e.g. Harlow and Welch (1965); Tackley and King (2003); Gerya and Yuen (2007)), because the latter has some advantages to grid-based advection methods, such as lack of numerical diffusion or dispersion and the possibility of representing sub-grid-scale features. All of the major geodynamical modelling codes include this option, including CitcomS (Moresi et al., 2014), Aspect (Heister et al., 2017), Stag3D/StagYY (Tackley and King, 2003; Tackley, 2008), TERRA (e.g. Panton et al., 2025), LaMEM (Kaus et al., 2016), and I3ELVIS (Gerya et al., 2015).

This method relies on many tracers (e.g. 5-50) being present in each cell. Thus, it is problematic that over time, gaps in the tracer distribution typically develop, often resulting in cells containing no tracers. At the same time, bunching of

tracers builds up. Such gaps and bunches typically develop when tracer advection does not respect the mass conservation equation, which can be due to (i) interpolation of velocities from grid points to tracer points not respecting conservation of mass (Pusok et al., 2016) and/or (ii) inaccurate advection of tracers in regions with large velocity gradients even with perfect velocities. The development of such gaps and bunches can be minimized by an optimal choice of tracer advection method (Pusok et al., 2016; Gerya et al., 2021) but apparently not eliminated, particularly since geodynamical simulations spanning the age of the Earth may require millions of time steps, giving small inaccuracies plenty of time to build up. Thus, some remedy, preferably one that is based on the equations being solved, is required. One remedy is to create new tracers to fill the gaps (Gerya, 2019), but this is unphysical.

Here, the mass conservation equation is used to correct non-conservation errors introduced during tracer advection by perturbing or "nudging" the positions of tracers. If performed frequently, such as every time step, the amplitudes of the nudges are small yet prevent large-scale gaps and bunches from building up. The method also works well when starting from an extremely uneven tracer distribution with large fractions of the domain initially devoid of tracers.

Irregularities in the spatial distribution of tracers can be quantified in terms of the number of tracers per unit volume (i.e. number density) calculated on a cell-by-cell basis. Alternatively, if tracers are considered to each have a mass (e.g. equal to the total mass of the domain divided by the number of tracers, although they could have different masses), then this can be thought of as a density, i.e. mass of tracers per unit volume. In the latter usage, the goal of this method is to nudge tracer positions in order to achieve, throughout the model domain, a tracer density that matches the correct physical density, which could vary with position if compressibility is included. The latter usage of "tracer density" is what the subsequent theory will mainly focus on.

In subsequent sections the mathematical theory is presented, followed by various tests of its effectiveness using the accompanying MATLAB program in two and three dimensions.

## 2 Mathematical Theory

50

As the goal is to achieve the correct tracer density everywhere, the first step is to calculate the current tracer density  $\rho_t(x,y,z)$  on a cell-by-cell basis. Tracer density can be defined either as the mass of tracers per unit volume or (if tracers are massless) the number of tracers per unit volume. It can be dimensional (kg/m³ or m⁻³, respectively) or nondimensional, as in the example MATLAB program. It is important that  $\rho_t$  changes smoothly as tracers move around, which it does not if one simply counts the number of tracers in each cell, because a tracer crossing a cell boundary causes an abrupt change in the densities of the two cells. Therefore, linear averaging of tracers to cells is important – termed "shape function" averaging by Tackley and King (2003) and widely recommended (e.g. Gerya, 2019; Ismail-Zadeh and Tackley, 2013). In this, each tracer contributes to the mass in 4 (in 2-D) or 8 (in 3-D) cells, linearly dependent on its distance from the cell centres using bilinear (in 2-D) or trilinear (in 3-D) functions

analogous to the shape functions used in the finite element method. Once the tracer-based density in each cell is known, the tracer density error can then be calculated as

$$\Delta \rho_e = \rho_t - \rho_c \ , \tag{1}$$

where  $\rho_c$  is the correct density (e.g. of rock). This can in general vary with position, making the method applicable without modification to compressible flows, but for the purposes of the tests in this paper  $\rho_c$  is assumed to be constant.

The required perturbation ("nudging") of tracer positions can be derived starting with the equation expressing conservation of mass:

$$\frac{\partial \rho}{\partial t} = -\nabla \cdot (\rho \vec{v}) \,, \tag{2}$$

where  $\rho$  is the density field,  $\vec{v}$  is the velocity field and t is time. Multiplying (2) by a finite time interval and substituting  $\Delta \vec{x} = \vec{v} \Delta t$  leads to an approximate equation relating a finite change in density to a finite perturbation in position  $\Delta \vec{x}$ , which is here applied to the tracer density  $\rho_t$ :

$$\Delta \rho_t \approx -\nabla \cdot (\rho_t \Delta \vec{x}) \,. \tag{3}$$

 $\Delta \vec{x}$  can conveniently be expressed as the gradient of a mass flux potential  $\phi$  (with units kg/m if  $\rho_t$  has units of kg/m<sup>3</sup> or m<sup>-1</sup> if has units of m<sup>-3</sup>):

$$\rho_t \Delta \vec{x} = \nabla \phi \quad . \tag{4}$$

Substituting (4) into (3) leads to a Poisson equation for  $\phi$ :

$$\Delta \rho_t = -\nabla^2 \phi \ . \tag{5}$$

The desired change in density  $\Delta \rho_t$  is minus the density error,  $\Delta \rho_e$ , therefore the equation to solve is

$$\nabla^2 \phi = \Delta \rho_e \ . \tag{6}$$

This can easily and efficiently be solved using standard methods such as multigrid. Assuming that the domain boundaries are impermeable, the appropriate boundary condition for  $\phi$  is zero gradient perpendicular to the boundary; for other velocity boundary conditions equation (4) can be used to derive the appropriate condition on  $\phi$ .

It is noted that another possible expression for  $\Delta \vec{x}$  is

$$\Delta \vec{x} = \nabla \varphi \tag{7}$$

where  $\varphi$  is a displacement potential (with units m<sup>2</sup>), leading to

$$\Delta \rho_t = \nabla \cdot (\rho_t \nabla \varphi) , \qquad (8)$$

which is slightly more difficult to solve and problematic in areas where  $\rho_t = 0$ , if such areas exist. Equation (4) also seems problematic in areas where  $\rho_t = 0$  but as there are no tracers in these areas, there is no problem in practice.

This method does not achieve a perfectly uniform tracer distribution in a single nudge because  $\rho_t$  changes (towards the correct density  $\rho_c$ ) during the displacement of tracers: equation (2) is an approximation. In areas of too-high  $\rho_t$  (decreasing during the correction step), equation (4) underpredicts the displacement, whereas in areas of too-low  $\rho_t$  (increasing during the correction step), equation (4) overpredicts the displacement. Thus, when calculating the displacement from equation (4) it is best to use an average of the initial density and the correct density, rather than only the initial density. Tests indicate that a geometric average gives slightly better convergence than an arithmetic average, but both perform considerably better than using just the starting  $\rho_t$ . In summary, when calculating displacement, equation (4) is replaced by:

$$\Delta \vec{x} = \frac{\nabla \phi}{\sqrt{(\rho_t \rho_c)}} \ . \tag{9}$$

A single application of this algorithm achieves a considerable reduction of the density error (quantified using the L1 or L2 norm), which is sufficient during a time-stepping situation. If, however, starting from an extremely non-uniform tracer distribution with large portions of the domain being devoid of tracers, several iterations of the algorithm may be needed, as documented in Section 4.

# 3 Accompanying MATLAB scripts






This method is implemented in two and three dimensions in the accompanying MATLAB scripts (Tackley and ETH Zurich, 2025) (main program NUDGE.m), which can run the various test cases documented and discussed in Section 4. MATLAB scripts have the advantage of being easy to translate into other science and engineering-oriented high-level languages that include multi-dimensional arrays and array algebra, such as Julia (Bezanson, 2017) or modern Fortran (Metcalf et al., 2024). Indeed, the method has already been implemented in the Fortran geodynamical modelling code StagYY (Tackley, 2008) and is in regular use.

The accompanying program uses a multigrid solver to obtain the displacement potential field. This is highly efficient but does require that the number of cells be a power-of-2 in each direction, or a power-of-2 times a small integer. Resolution is set by the number of cells in each direction nx, ny and nz, and the number of tracers by  $tracers\_per\_cell$ . Two-dimensional cases can be run by setting the number of y-points ny=1. Densities are calculated at cell centres, while displacements and velocities are defined at cell boundaries in the standard staggered-grid finite volume arrangement (e.g. Harlow and Welch 1965; Patankar, 1980) as used by many codes in the geodynamical modelling community (e.g. Ogawa et al., 1991; Tackley, 1993; Trompert and Hansen, 1996; Gerya and Yuen, 2007; Kameyama et al. 2008; Tackley, 2008; Kaus et al., 2016). Domain boundaries are coincident with the perpendicular displacement/velocity points. Tracer positions are initialised either on a regular grid (with a smaller grid spacing than that on which the velocities/displacements are calculated), on a regular grid with random perturbations of up to half a grid spacing, or completely randomly. Initialising

tracers on a regular grid causes artefacts with tracer alignment when they are advected, so regular + random is optimal. Completely random positions cause a density error that is typically a factor of 2 larger than regular + random, as shown later.

The domain depth is assumed to be 1.0 and the grid spacing is the same in all three physical directions, meaning that the domain width in the x and y directions is given by (nx/nz) and (ny/nz), respectively.

The MATLAB m-files are:

- NUDGE.m: The main program that runs and plots individual tests or test suites.
- correct tracer density.m: Performs the "nudging" algorithm detailed in Section 2.
- tracer density.m: Calculates the cell-based tracer density field.
- Poisson solve.m: Solves Poisson equation in 2-D or 3-D assuming zero-gradient boundary conditions.
- advect tracers.m: Performs 1st-order Euler, 2nd-order or 4th-order Runge-Kutta tracer advection.

The core of the nudging algorithm in correct tracer density m is compact, consisting of only four lines (Fig. 1).

Figure 1. MATLAB implementation of the algorithm in Section 2, in file correct\_tracer\_density.m

#### 4 Tests






Four test cases are presented. The first starts with various extremely non-uniform tracer distributions and tests how rapidly (in terms of number of nudging iterations) the method can create a uniform tracer distribution. The other three test cases involve time stepping, with the first two of these using analytical flow fields (cellular flow and shear flow along an interface) but the third being full thermal convection. After these, timings of the various routines are presented. Finally, adaptive use of the nudge correction (i.e. using when needed rather than once every time step) is tested.

# 4.1 Highly non-uniform tracer distribution tests

Various idealized initial tracer distributions are tested:

- (i) Half-empty. Half of the domain is empty of tracers.
- (ii) Rectangular hole. A rectangular region in the middle of the domain is empty of tracers.
- (iii) Spherical hole. A spherical region in the centre of the domain is empty of tracers.
- (iv) Sphere. All tracers are in a sphere in the centre of the domain, the rest being empty.
- (v) Random. Tracers are placed randomly in the entire domain.

Figure 2 (top row) shows these initial conditions and Fig. 2 (rows 2-5) shows the results of the first four correction nudges. After two nudges (a "nudge-nudge"; Fig. 2 middle row) tracers fill the domain; the subsequent nudge-nudge evens them out further. The evenness of the tracer distribution is quantified by tracer density plots in Fig. 3. After one nudge-nudge there is

still significant unevenness, but this becomes difficult to discern after a further nudge-nudge. Random initial tracer positions (right column) lead to substantial initial unevenness in tracer density.




Figure 4 shows how the L1 norm of tracer density error decreases with number of nudges for the 2-D tests (Figs. 2 and 3) and for 3-D versions of the tests. For highly non-uniform initial conditions the reduction in tracer density error is more than an order of magnitude after 2 nudges, then becomes less rapid. Again, the random initial condition has substantial tracer density error approaching 0.2. 3-D cases are similar but with slightly slower convergence for the "sphere" case.

A problem in initial tests of the "sphere" case was that many tracers were nudged through the domain boundaries. This is due to the extreme nature of this test and is not a problem in a normal time-stepping application, but nevertheless a solution has been found. An approach that does not work is to place these tracers at the closest point inside the domain, although this does work for normal tracer advection by a velocity field that does not cross the boundaries. However, in this application the displacement field can substantially cross the boundaries, leading to a build-up of tracers at the boundaries, tracers that are not easily nudged away from there (close to the boundaries the perpendicular displacement is 0). What does work is to detect tracers that are nudged beyond external boundaries and instead apply only a fraction of the displacement to them. A fraction of 70% was found to be optimal. That is, tracers that are initially calculated as crossing boundaries are instead moved only 70% of the calculated distance.

Figure 2. Tracer positions in the five highly nonuniform tests performed in 2-D with 32x32 cells and 10 tracers per cell on average. Each column is one test case and each blue dot is a tracer. Shown are (top row) the initial condition and (rows 2 - 5) nudges 1-4.

Figure 3. Tracer density error fields for the tracer distributions shown in Figure 2. The colour bar is the same for all frames.

Figure 4. L1 norm of tracer density error versus number of nudges for the 5 initial tracer distributions in (left) 2-D 32x32 cells and (right) 3-D 32x32x32 cells, in both cases with 10 tracers per cell on average.

# 4.2 Time-stepping cellular-flow test



The goal in this test is to determine whether the tracer nudging method can prevent gaps and bunches from building up in a time-stepping situation, as this is what is typically used in geodynamical simulations. Tracers are advected according to an analytically defined velocity field given by the curl of a two-dimensional stream function S(x,z):

$$v_x = \frac{\partial S}{\partial z}$$
  $v_y = 0$   $v_z = -\frac{\partial S}{\partial x}$  . (10)

The resulting flow field is divergence-free for any S. In the presented tests, S is defined by

$$S(x,z) = \frac{1}{\pi} \sin\left(\pi \frac{x}{L_x}\right) \sin\left(\pi \frac{z}{L_z}\right)$$
 (11)

where  $L_x$  is the length of the domain in the x-direction and  $L_z$  is the length of the domain in the z-direction. This gives a one-cell circulation pattern with no flow through the boundaries and velocities given by:

$$v_{x} = \frac{1}{L_{x}} \sin\left(\pi \frac{x}{L_{x}}\right) \cos\left(\pi \frac{z}{L_{z}}\right) \qquad v_{z} = -\frac{1}{L_{x}} \cos\left(\pi \frac{x}{L_{x}}\right) \sin\left(\pi \frac{z}{L_{z}}\right)$$
(12)

In order to maximize the challenge of maintaining a uniform tracer distribution, tracers are advected using the first order forward Euler method, which usually makes them spiral outwards and concentrate towards the outside of the domain. This combination (Euler advection, 1 nudge per time step) is compared to three advection methods without any nudging: Euler, 2<sup>nd</sup>-oder Runge-Kutta and 4<sup>th</sup>-order Runge-Kutta methods. Velocities at the staggered grid points are calculated using Equation (12) and linearly interpolated to tracer positions. Tracers are initialized on a (regular+random) grid as discussed

earlier, except in an additional (Euler advection, 1 nudge per time step) case with tracers initialized in completely random positions, to test what difference that makes.





Figure 5 shows tracer distributions and density error fields after 100 time-steps of nondimensional time 0.05 on a 32x32 grid with an average of 10 tracers per cell. As the maximum velocity given by equation (12) is 1, tracers move a maximum distance of 0.05 in one step. As expected, the Euler method (1st column) is quite inaccurate, with tracers spiraling outwards and building up towards the domain boundaries and corners. With the addition of a single nudge per step (right two columns), however, the tracer distribution remains even and negligible tracer density error is visible. The 2nd- and 4th-order Runge-Kutta methods produce similar results to each other, with significant unevenness visible in the tracer density error field.

Figure 5. Tracer distributions (top row) and associated density error fields (bottom row) for the 4 advection methods on a 32x32 grid with an average of 10 tracers per cell. The right-most combination (Euler advection plus nudge correction) is performed with both completely random initial tracer positions (4th column) as well as the default positions. The colour bar is the same for all density error fields.

The time-evolution of tracer density error is quantified in Fig. 6, which shows the L1-norm versus time step. The "Euler" case rapidly develops a large density error, whereas in both Runge-Kutta cases the error increases steadily from the initial condition, surprisingly at a similar rate for the 2nd- and 4th-order schemes. Adding a single nudge per step to Euler advection causes a reduction of density error to a low value, which is subsequently maintained. A completely random initial condition has a density error of ~a factor of 4 higher than (even + random), indicating that the latter initial condition is much better. Even so, adding a nudge correction per step rapidly reduces the error.

Figure 6. L1-norm of tracer density error versus time step for the tests in Figure 5.

# 210 4.3. Opposing flow on an interface test

Pusok et al. (2017) thoroughly tested many marker advection methods using four different tests, of which arguably the most challenging was the first one, in which two rigid blocks move in opposite directions along an interface oriented at 45 degrees to the grid, mimicing to a subduction interface, for example. Material above the interface has a velocity (vx, vz) = (1,1) while material below the interface has a velocity of (-1,-1), thus creating a large (shear) velocity change over one grid spacing.

Tracers advected out of the domain are wrapped around at the appropriate place on a 45 degree line from where they left the domain. The advection methods tested in Pusok et al. (2017) almost all created a gap along the interface. It is here tested whether the nudge correction can avoid the gap along the interface.

Figure 7. Tracer positions (blue) and tracer density error fields for the opposing flow on an interface test after 500 time steps and a 32x32 grid. See Pusok et al. (2017) for details of this test and comparisons with additional advection methods.

Figure 7 shows the tracers and tracer density error fields using Euler, Runge-Kutta 2nd-order or Runge-Kutta advection, either on their own or with a single nudge correction per time step. Euler or 2nd-order Runge-Kutta methods indeed create a gap, while with the 4th-order Runge-Kutta method there is a band of tracers inside the gap. Away from the interface, the initial relative positions of tracers are preserved.



With one nudge correction each time step, tracer density error maps are greatly improved by eliminating the gap as well as reducing errors away from the gap. However, examination of tracer positions does show artifacts around the interface. The Euler and 2nd-order Runge-Kutta cases now display a series of small gaps oriented at 45 degrees instead of one big gap. These are sub-grid-scale features that do not affect the cell-based tracer density field and so are not eliminated by tracer nudging. It could be that the rather artificial 45 degree angle of the interface allows these features to persist. The 4th-order Runge-Kutta test is much better. This is the only test in this study in which the 4th-order Runge-Kutta shows a distinct advantage over 2nd-order Runge-Kutta.

Graphs of tracer density error vs. time for advection (Figure 8) show a rapid increase in error to begin with, subsequently stabilising and increasing only slowly. In the nudge-corrected cases, the error decreases then stabilises at a value roughly an order of magnitude lower than that of the uncorrected cases.

Figure 8. L1 norm of the tracer density error versus time step for the tests in Figure 7.

## 240 4.4. Thermal convection test



The final test setup is that of thermal convection in an infinite Prandtl number fluid with strongly temperature-dependent viscosity and is thus representative of an actual geodynamical simulation. The Boussinesq approximation is assumed, with the fluid heated from below (nondimensional temperature T=1.0), cooled from above (T=0), having no internal heating and an exponentially varying viscosity  $\eta(T) = exp[-13.8155(T-0.5)]$ , which gives a factor of  $10^6$  viscosity variation. The Rayleigh number (at T=0.5) is  $10^6$  and the mechanical boundary conditions are all free slip. The initial condition has

$$T(x) = 0.5 + 0.01\sin\left(\pi\left(\frac{1}{2} + 3\frac{x}{L_x}\right)\right),\tag{13}$$

which leads to the formation of two hot plumes from the lower thermal boundary layer, as shown in Figure 9. Flow is more rapid in these plumes due to their low viscosity. The time step is limited by the Courant condition because a finite-volume scheme is used for temperature advection and for thermal diffusion. The test is run for 500 time steps on a 32x32 grid with an average of 10 tracers per cell.

Figure 9. Non-dimensional temperature and viscosity fields for the thermal convection test after 500 time steps. They are the same regardless of tracer advection method because tracers are purely passive.

Tracer positions and density errors (Figure 10) show large artifacts for the Euler advection method, which are much reduced by using Runge-Kutta advection. With one nudge iteration per time step, tracer density errors are greatly reduced for all three advection schemes.

Figure 10. Tracer distributions (top row) and associated density error fields (bottom row) for the three advection methods without (left 3 columns) or with (right 3 columns) a nudge correction each time step.

The L1-norm of density error (Figure 11) shows a rapid and continuing increase for the Euler scheme, but a much less rapid increase for the Runge-Kutta schemes. 4th order and 2nd order schemes give almost the same result, as is also visible by comparing the tracer distributions in Figure 10. With a single nudge iteration per step, tracer density error is reduced to a much lower value than that of the initial condition, where it remains stable at around an order of magnitude lower than the Runge-Kutta advection schemes. The error has the same magnitude for Euler and Runge-Kutta schemes. Thus, for pure


advection there is no advantage to using 4th order instead of 2nd order, while with a nudge correction each step there is no advantage to using Runge-Kutta instead of Euler.

Figure 11. L1 norm of the tracer density error versus time step for the tests in Figure 10. The RK 4th order curves mostly overlie the RK 2nd order curves, which are therefore not visible.

## 4.5 Timing analysis



|                 | Euler | RK2   | RK4   | Nudge | Stokes solve |
|-----------------|-------|-------|-------|-------|--------------|
| 32x 32 10/cell  | 43.08 | 69.35 | 124.2 | 50.66 | 16.4         |
| 128x128 20/cell | 1187  | 2057  | 3935  | 1384  | 360.1        |

Table 1. For the thermal convection test, timings (in milliseconds) for first-order Euler tracer advection, 2nd- and 4th-order Runge-Kutta tracer advection, a nudge correction, and a 2-D Stokes (v,p) solve, at two different resolutions: 32x32 cells with 10 tracers/cell and 128x128 cells with 20 tracers/cell. Measured on a single core of a 3.8 GHz Intel Core i5 in a 2017 iMac, averaged over 100 time steps.

Table 1 lists the CPU time taken for various order tracer advection steps, compared to one nudge correction and a 2D Stokes (v,p) solve. The increase in CPU time from 1st order (Euler) to 2nd-order Runge-Kutta to 4th-order Runge-Kutta is notable, with the latter taking about three times as long as first-order Euler. A nudge correction takes slightly more CPU time than an

Euler advection step, indicating that moving the tracers dominates the time; calculating the displacement field is relatively fast. The Stokes solve (see Matlab script direct\_solve\_Stokes\_2D.m) solves for (vx, vy, p) on a staggered grid using Matlab's built-in "\" sparse direct solver, which uses UMFPACK. While the Stokes solve here seems fast compared to tracer advection, it is important to note that this is using a compiled, highly optimised solver while the tracer routines here use interpreted Matlab - if implemented in a compiled language like C, Fortran or Julia than they would likely be much faster, while the Stokes solve would not be.

Comparing the two different resolutions indicates that, as expected, the time taken for tracer operations scales in proportion to the number of tracers: the higher resolution has 32 times as many tracers and requires proportionally more time. In contrast (also as expected) the time required for the Stokes solve increases more rapidly than the number of unknowns: the higher resolution has 16 times as many unknowns but takes 22 times longer.

## 4.6 Adaptive nudging





Instead of making one nudge correction every time step, another idea is to specify the required level of accuracy (in terms of L1 norm of tracer density error) and make a correction only when needed, or multiple iterations per step if a particularly low error is desired. This approach has been tested using the cellular advection test and the convection test, with results listed in Table 2. The error of  $3.5 \times 10^{-2}$  corresponds to the tracer density error associated with the initial condition (i.e., tracers initialised on a grid plus random perturbations), which therefore seems like a reasonable value to stay below. For the cellular advection test with this choice, a nudge needs to be made in 82% of time steps with 1st order (Euler) advection, dropping to 70% then 67% for 2nd- and 4th-order Runge Kutta, respectively. If the error criterion is relaxed to  $5.0 \times 10^{-2}$  then the required number of nudges drops considerably (38%, 25%, 25%), whereas if it is made stricter at  $2.0 \times 10^{-2}$ , then almost three nudges are required per time step regardless of the advection method.

|                                                               | Euler        | RK2          | RK4               |  |  |  |
|---------------------------------------------------------------|--------------|--------------|-------------------|--|--|--|
| Cell; dt=0.05; 100 steps; error=5.0x10 <sup>-2</sup>          |              |              |                   |  |  |  |
| Nudges/step                                                   | 38%          | 25%          | 25%               |  |  |  |
| t <sub>total</sub> (s)                                        | 6.298 8.218  |              | 13.48             |  |  |  |
| Cell; dt=0.05; 100 steps; error=3.5x10 <sup>-2</sup>          |              |              |                   |  |  |  |
| Nudges/step                                                   | 82%          | 70%          | 67%               |  |  |  |
| t <sub>total</sub> (s)                                        | 8.614        | 11.13        | 15.94             |  |  |  |
| Cell; dt=0.05                                                 | 5; 100 steps | ; error=2.02 | x10 <sup>-2</sup> |  |  |  |
| Nudges/step                                                   | 292%         | 291%         | 291%              |  |  |  |
| t <sub>total</sub> (s)                                        | 19.48        | 22.07        | 29.86             |  |  |  |
| Convection; dt=Courant; 500 steps; error=3.5x10 <sup>-2</sup> |              |              |                   |  |  |  |
| Nudges/step                                                   | 14.8%        | 13.8%        | 13.8%             |  |  |  |

| t <sub>total</sub> (s) | 7.170 | 11.02 | 18.47 | 1 |
|------------------------|-------|-------|-------|---|
|------------------------|-------|-------|-------|---|

Table 2. Average number of nudge corrections per time step required to keep tracer density error below a specified threshold, and total execution time of tracer operations for all time steps (in seconds), for the cell test with three different accuracies and the convection test with one accuracy. Timed on an Apple M4 Pro CPU (MacBook Pro Nov 2024).

Thus, there is a trade-off between increasing accuracy of advection and decreasing frequency of needed nudge corrections. In terms of execution time, however, it is in every case fastest overall to use the lowest advection accuracy (Euler) with somewhat more frequent nudge corrections. The increase in advection accuracy from 2nd-order to 4th-order Runge-Kutta is not justified, because the total execution time rises considerably but the frequency of nudges remains almost the same.

In the convection test (Table 2 lowest section), nudges are needed considerably less often (in 14.8-13.8% of steps) than in the cellular advection test with the same error criterion of  $3.5 \times 10^{-2}$ , despite the increased complexity of the flow. This is because the time step is smaller in the convection test: it is limited by the Courant condition such that the advection distance is a maximum of half a grid spacing, whereas in the cell test there is no such limitation and a time step of 0.05 is used, during which tracers may be advected several grid spacings (up to 1.6 for nz=32).

In summary, the results of these tests indicate that significant execution time can be saved by taking a nudge only when needed, and that the fastest approach is to use first-order advection, even though nudge corrections are needed slightly more frequently.

#### 5. Conclusions







The tracer nudging method presented here is an effective way of eliminating and preventing gaps and bunching of tracers in geodynamical models/simulations. It uses the mass conservation equation to calculate tracer position perturbations ("nudges") that correct mass conservation errors introduced by tracer advection. Starting from an extremely heterogeneous distribution with large regions of the domain devoid of tracers, it can produce an even distribution in only a few nudge iterations. In a time-stepping situation it allows a low-order tracer advection method to be used while maintaining a tracer distribution that is more even than that obtained using high-order advection methods alone. The computational cost is small and dominated by performing a first-order tracer advection operation, because the other part simply involves solving a Poisson equation. A nudge correction may not be needed every time step, which further reduces computational cost. It is more time-efficient to use nudge corrections in conjunction with low order tracer advection rather than high order tracer advection, even though the latter reduces the needed frequency of corrections.

**Code availability.** The exact version of the MATLAB code used to produce the results and figures used in this paper is archived on Zenodo under the MIT license under DOI 10.5281/zenodo.17058894 (Tackley and ETH Zurich, 2025). No input data or additional scripts are required.

Data availability. There is no data, only software code. The data plotted in the figures can be generated by running the code.

**Competing interests.** The author declares that he has no conflict of interest.

Special issue statement. Advances in numerical modelling of geological processes

Acknowledgments. The author is grateful to ETH Zurich. The author was motivated to write this method up by discussions on tracer advection with Boris Kaus, Thibault Duretz and Taras Gerya. The author thanks two anonymous reviewers for thoughtful suggestions that improved the manuscript.

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
