# Peer review of "The tracer nudging method for correcting and preventing uneven tracer distributions in geodynamical models"

_EGUsphere, 2025_

## Author Comment (AC1)

Following instructions, in this Author Comment (AC) I respond to the two Referee Comments (RCs) without submitting a revised manuscript. This is the first time I have encountered such a review process, as for most journals the response and revised manuscript accompany each other. As a result, this AC is addressing/explaining any conceptual questions and broad recommendations that were raised in the RCs, but not presenting any new tests or responding to line-by-line rewording recommendations, as these are things to do in the context of a revised manuscript.

The two RCs are extremely helpful in pointing out clarifications, further explanations and further tests that would make the manuscript more helpful to the community, so I certainly thank the referees for their efforts.

There are two recommendations in common: to include a test or tests that directly resemble geodynamic problems, and to examine the trade-off in execution time between this method versus high-accuracy advection methods, so I would certainly do this in a revised manuscript. As a comment, the two existing test cases are arguably more challenging than a typical geodynamic application as the first one has a much more extreme non-uniform tracer distribution than ever encountered while the second one uses a much worse advection method than normally used.

-------------------------------------------------------- RC2 --------------------------------------------------

This manuscript tackles an important numerical difficulty in geodynamic modelling: the development of uneven tracer distributions during Lagrangian particle advection. The authors propose a *tracer-nudging* algorithm, derived from the requirement that material density remain constant, which iteratively redistributes tracers until a uniform spatial density is achieved. The idea is elegant and, if widely adopted, could mitigate one of the longest-standing practical problems in high-resolution mantle-convection and lithosphere-deformation studies. While I acknowledge the novelty and potential impact of the work, several aspects of the presentation and validation need to be strengthened before the paper is suitable for publication.

**Major comments**

1. **Demonstrate the method on realistic geodynamic problems**

   The manuscript shows simple circulation tests only. Please include at least one geologically meaningful application—e.g. a high-viscosity-contrast convection benchmark or a 2-D subduction experiment—to illustrate how tracer nudging behaves in complex, time-dependent flow fields and when and how often the nudging is needed.

   2. **Compare with established schemes**

The paper states that the computational cost of tracer nudging is small, yet no comparison is provided. As far as I can see from the manuscript, the additional computational cost is probably higher than the existing remedies that only add a correction items to the velocity interpolation. It would be nice if the author make a comparison with other method and emphasize the advantages (and limitations) of this method

I would certainly incorporate 1 & 2 in a revised manuscript. A few comments below regarding what I would expect.

Regarding 1, as noted above the two existing test cases are arguably more challenging than a typical geodynamic application. I like the 4 tests in Pusok et al. (2016) as possible additional ones to add as they are well documented and many marker advection schemes are already tested using them. It is notable that their more idealised tests (sharp interface and SolCx) present great difficulties for the tested advection schemes, with most failing within a few 100 time steps.

Regarding 2, I am not aware of any other scheme that can preserve an even tracer distribution for an indefinite number of time steps: they all start failing at some point. As noted above, some previously-tested marker advection schemes fail simple tests in a few 100 time steps. In contrast, age-of-Earth global geodynamical simulations often require of order (1 million) time steps. One expects a trade-off between the accuracy of the advection scheme and how frequently a nudge correction is needed, and it would be interesting to quantify this.

**3. Clarify applicability to compressible versus incompressible flow**

Lines 34–35 imply the method corrects non-divergence-free advection errors. However, many geodynamic models employ *compressible* Stokes flow, where $\nabla \cdot \mathbf{v} \neq 0$ by definition.

1.
   o State explicitly that the current formulation targets incompressible Stokes problems.

   o Discuss whether, and how, the nudging algorithm could be adapted to compressible flow, and what errors might arise if it were used without modification.

   o

The algorithm works for compressible flows without modification. This is because $\rho_c$ in Equation (1) can be an arbitary function of position (which was already stated, but the earlier sentence that the referee pointed out implies something different). So I would explain that better in a revised manuscript.

**4. Streamline the figures**

Figures 2 and 3 convey nearly the same information. I suggest keeping the one that best illustrates the tracer-density evolution and moving the other to the supplement or removing it entirely.

Figures 2 and 3 are really complementary because by comparing the two figures the reader can see clearly how a tracer distribution converts to a tracer density field (something that is not normally plotted in publications and therefore unfamiliar to the readers). So I would argue that it's best to keep both close to each other in the main text.

-------------------------------------------------------- RC2 --------------------------------------------------

The study by P. Tackley presents a new algorithm for tracer advection for the marker-in-cell method. The algorithm corrects the displacement of tracers by solving for mass conservation of tracers. It is a neat, physics-based approach; however, I find that the choice of the potential function requires more discussion, and, if this algorithm is to be useful to the geodynamics community—or more broadly, to anyone using the marker-in-cell method, the writing can be improved and more tests would be useful. I detail these points below. Considering that the revision work I propose is considerable, I recommend a major revision of the manuscript.

1. Choice of the potential function for displacement vector in Eq. 3.

It is not clear why the author chose Eq. 4 (instead of Eq. 7) besides convenience to solve a linear Poisson equation. The justification is not there. I agree that it is more convenient to define dx=grad(phi)/rho_t, but what is the physical meaning of this equation? What would be the physical units of such a displacement? Maybe the author has thought about this, but the displacement should not depend on tracer density, and only Eq. 7 is a physical choice. Neither Eq. 4 nor Eq. 9 is balanced in terms of units. Has the author tried solving the non-linear Poisson equation in Eq. 8?

The meaning of Equation (4) is clearer if written as

$$\rho_t \Delta \vec{x} = \nabla \phi$$

The left-hand side is mass flux, thus $\phi$ is a mass flux potential (I would rename it in the text). As the continuity equation is about div(mass_flux) and this correction algorithm is about redistributing mass (to acheive the correct density), it is logical and physical to use a mass flux potential. The (SI) unit of displacement is metres and the units of mass flux above are kg/m$^2$, the physical meaning being kilograms flowing through a 1 m$^2$ area in one correction step. (In the continuity equation, mass flux is per second rather than per correction step). The units of $\phi$ are kg/m and equations (4) and (9) are dimensionally consistent with this.

The alternative definition of $\phi$ used in equations (7) and (8) is a displacement potential with units of m$^2$. I am considering using a different symbol for this to avoid confusion.

The example code is nondimensional so there are no units.

L77, Eqs. 7-8: Why is the non-linear equation problematic in areas where rho_t is zero? Not sure this is an issue because in the marker-in-cell method, zero tracer density in control volume is a violation, so this is avoided (which makes the models in Fig. 2 somewhat artificial).

Discretization of partial differential equations such as (8) generally leads to a matrix problem of the form:

      [coefficients][unkowns] = [rhs]

where [coefficients] is a square matrix, [unknowns] is a column vector containing the quantity being solved for at each grid point (in this case $\phi$) and [rhs] is a column vector of the know fixed

right-hand-side. If too many coefficients are 0 then the matrix is singular and thus no solution is possible. That is the case here. The presence of rho_t=0 in one or more cells results in a singular matrix and thus no solution.

Yes in principle the marker-in-cell method should have at least one marker in each cell, something that this method is intended to enforce. But after an advection step, it can be that there are cell(s) with no tracers and therefore this method would not work.

Minor comments regarding mathematical theory:

L46, Eq. 1: Need definition (equation) for tracer density in the main text. It was obtained from the code but was not obvious. Density = #nlocal_mark * nx*ny*nz / #ntracers

Equally, for the mean density, which is taken equal to 1 here.

Mean density = nmarkx_cell*nmarkz_cell * nx*ny*nz / #ntracers = 1

It was already stated that tracer density is mass of tracers per unit volume. Depending on the application, this could be nondimensional mass and density (as in the example code, this is why the mean value is 1.0) or dimensional mass and density (kg/m$^3$), in which case the equations pasted by the referee would be different.  I would put some further explanation and define this better.

L47: tracer density is "defined as the mass (or number) of tracers per unit volume". This statement is correct only if it is assumed that all tracers have the same mass, and the density relates to the number of tracers. This needs to be clarified.

People use the marker-in-cell approach in different ways. Sometimes markers do not have mass, in which case the "number density" is the relevant quantity, and this is what Pusok et al. (2016) plot, for example, actually they used number per cell rather than number per volume. Sometimes markers do have mass, in which case the usual "mass density" is the relevant quantity. In the latter case, markers sometimes have different masses. The method works for all of these possibilities without changing the equations in the manuscript. The only difference would be that the units of $\phi$ are different if using "number density" instead of "mass density", as I would plan to point out in a revised manuscript.

Eq. 2 to Eq. 3: the discrete version is missing for v=dx/dt.

 Most likely the reader can follow this step, but I could point this out just in case.

L55-57, Eq. 1: rho_c is mean density of tracers in this study. To generalise, better to use 'initial' instead of 'correct' (i.e., rho_0). Also for the purpose of this study, rho (without the t subscript) can be used for the tracer density.

In general, rho_c can vary with position (see response to RC1 above) and can also change with time - it does not have to be the initial or the mean density. Therefore it is best to leave the subscripting as is and for me to add some explanation.

2. Generalisation and stopping criterion for nudging

L111: would the equations change for variable grid spacing?

No change. Equations (1)-(9) are physical so there is no change to them when the grid spacing varies (just as there is no change to the continuity, Stokes or energy equations).

If the numerical implementation uses linear shape functions to calculate tracer density (as the example code does), then they of course are adapted to the grid spacing and so will be sampling a smaller or larger volume depending on the grid spacing, as is standard when averaging any tracer-based quantity to the grid.

Also, is the scheme generalisable for other boundary conditions, other than impermeable (L72)? Can the boundary conditions for Poisson be generalised from the velocity boundary conditions?

Yes. Given some equation(s) for the velocity boundary condition, equation (4) can be used to derive the relevant equation(s) for $\phi$.

The number of 'nudges' required seems arbitrary and it is not clear what is the computational cost of each nudge. I wonder if a stopping criterion can be derived? Example, norm(density error)<=tolerance, where tolerance = 1e-1 or 1e-2 from Fig. 4. Is that an overkill for using the algorithm?

A good idea, as a nudge might not be necessary every time step if the advection scheme is accurate enough. As people don't normally complain about the initial tracer distribution in geodynamic models, perhaps a good tolerance to use is the initial L1-norm of the tracer density error, which for the second test is about 3.5e-2 (see Figure 6).

3. Geodynamic and performance tests

The tests presented are highly simplified, and maybe artificial (Fig. 2). To demonstrate better the impact of the algorithm, I recommend adding another test showing a geodynamic problem— with sharp velocity interfaces or rotational flows near corners. It would be interesting to see, for example, the problem cited in L27 on eruption/intrusion, which was stated, but not explained, and how this algorithm deals with it.

See response to RC1 - I certainly plan to add one or more tests.

Then, there is the issue of the computational cost of this algorithm (with nudges). Simple Matlab scripts run fast, but the marker-in-cell method becomes expensive for 2-D and 3-D problems. What is the cost of this algorithm? How does it vary with grid size (resolution)?

L201-202: says the computational cost of solving Poisson is small, but need to demonstrate. For example, using a geodynamics test case, show time to solve for Stokes and time for tracer correction (Poisson*number_nudges).

See response to RC1 - I certainly plan to add one or more tests of the computational time aspect.

4. Focus and style of writing

The algorithm corrects the displacement of markers by solving for conservation of tracer mass. This does not transpire from the nudging and bunching described. The writing is too focused on the algorithm (which comes out almost as a trick to deal with marker dispersion/clustering), and not so much on the physics or why is it better than other methods. I think it is not doing a great job advertising why this could be a useful technique for geodynamics, and the informal style of writing does not help (e.g. usage of regular+, Euler+nudge+random without clear definition).

Other examples,

L40: the goal shouldn't be to nudge the tracers, but to correct their displacement in order to keep uniform density.

L46: the goal is to preserve the initial tracer distribution.

The Abstract should also be revised because I had the following questions:

L9: is the correction method physics-based? A sentence describing the method is lacking. Only results are described.

L14: what does the author refer to as non-conservation errors? How do they occur? They are introduced somewhat late, even in the abstract.

I'll certainly consider these suggested rewordings/clarifications if preparing a revised manuscript.

Non-conservation errors refers to tracer advection not perfectly obeying the mass conservation equation (2).

The details on the content of the Matlab routines in Section 3 (L112-118) and Fig 1. should be in an appendix. Instead, the author can outline a pseudo-algorithm such as

- Initialise tracer distribution, rho = rho_0
- Time loop:
- advect tracers with velocity field v,
- calculate rho and density error,

- correct tracer location by solving for displacement dx (Poisson equation).

I would like to present the code in whatever manner is normal and customary in such papers so will seek clarification from the journal staff/editors as to what that is.

Minor comments

It's great that the referee has spent time writing so many minor suggestions for improving the wording and adding explanations. These are things to address in the context of a revised manuscript; I don't see anything that needs clarifying here.

L6-7, 17: tracers may track other fields besides composition and temperature throughout the domain. Their advantage is that they may perform better at advecting these fields compared to grid-based methods.

L7: why does the problem occur? i.e., rotating fields in a box domain or at sharp interfaces.

L8: Suggest to use 'clustering' or 'accumulation' instead of 'bunching'.

L20: (e.g., \cite not \citep)

L21: why all the major codes use marker-in-cell? It is better to justify advantages rather than state a common practice.

L24: (e.g. 5-50) not necessary. The accuracy of the marker in cell technique relies on having a high density of tracers in each cell.

L29: why the errors occur in the first place?

L30: physics-based remedy is required

L31: one previous solution is to create new tracers, but why is this not a good remedy?

L44, L96: just because it is already implemented in StagYY, it is not a statement of validation or verification. StagYY should cite the work in the current manuscript to demonstrate the method, not the other way around.

L109: first time density bounds are given; this should occur earlier when density is defined.

L149: unclear statement. Tracers at the boundaries were nudged only a fraction of the calculated displacement to avoid crossing the boundaries.

Eqs. 11-12: could plot either the stream function or plot the velocity field to understand why the center particles might get dispersed.

Section 4.2 notation: need partial differentials (\partial).

Eq. 12: mistake Vx = 1/Lz sin() cos()

L172: define Euler+nudge+random

L174: calculated as in Eq. 12 (not above)

Figures 2 and 3 do not have colorbars or state that blue points represent tracers.

Fig 5: what is the computational cost or runtime between the cases? Add colorbars.

Fig 6: all cases should start with the same initial condition, and then additionally a case with completely random distribution.

Matlab: Add in readme how to run the script. I.e. change ntest = 1; choice between 1-11.

---

## Author Response (AR1)

The two RCs are extremely helpful in pointing out clarifications, further explanations and further tests that would make the manuscript more helpful to the community, so I certainly thank the referees for their efforts.

There are two recommendations in common to both RCs: to include a test or tests that directly resemble geodynamic problems, and to examine the trade-off in execution time between this method versus high-accuracy advection methods, so I have added two more tests and studied the CPU time trade-off.

This manuscript tackles an important numerical difficulty in geodynamic modelling: the development of uneven tracer distributions during Lagrangian particle advection. The authors propose a *tracer-nudging* algorithm, derived from the requirement that material density remain constant, which iteratively redistributes tracers until a uniform spatial density is achieved. The idea is elegant and, if widely adopted, could mitigate one of the longest-standing practical problems in high-resolution mantle-convection and lithosphere-deformation studies. While I acknowledge the novelty and potential impact of the work, several aspects of the presentation and validation need to be strengthened before the paper is suitable for publication.

**Major comments**

**1. Demonstrate the method on realistic geodynamic problems**

The manuscript shows simple circulation tests only. Please include at least one geologically meaningful application—e.g. a high-viscosity-contrast convection benchmark or a 2-D subduction experiment—to illustrate how tracer nudging behaves in complex, time-dependent flow fields and when and how often the nudging is needed.

I have added two more tests, one of which is a "realistic geodynamic problem", namely high-viscosity-contrast thermal convection.

**2. Compare with established schemes**

The paper states that the computational cost of tracer nudging is small, yet no comparison is provided. As far as I can see from the manuscript, the additional computational cost is probably higher than the existing remedies that only add a correction items to the velocity interpolation. It would be nice if the author make a comparison with other method and emphasize the advantages (and limitations) of this method

I have added a timing analysis (new Section 4.5). This shows that a nudge correction takes slightly more CPU time than performing first-order tracer advection. Since the correction procedure includes performing first-order tracer advection, it means that calculating the displacements takes a relatively short time. Examining the trade-off between order of advection (1st, 2nd or 4th) and how frequently a nudge correction is made to keep tracer density error lower than its original value indicates that optimal CPU time efficiency is obtained using 1st order advection with relatively more frequent nudges, rather than using higher order advection with less frequent nudges (new Section 4.6).

I am not aware of any other scheme that can reduce the error in tracer density (distribution). It's usually a question of minimising how rapidly things get worse; they all start failing at some point. Age-of-Earth global geodynamical simulations often require of order (1 million) time steps making it virtually impossible to find an uncorrected scheme that can prevent substantial degredation in tracer distribution.

**3. Clarify applicability to compressible versus incompressible flow**

Lines 34–35 imply the method corrects non-divergence-free advection errors. However, many geodynamic models employ *compressible* Stokes flow, where  $\nabla \cdot \mathbf{v} \neq 0$  by definition.

- 1.
- State explicitly that the current formulation targets incompressible Stokes problems.
- Discuss whether, and how, the nudging algorithm could be adapted to compressible flow, and what errors might arise if it were used without modification.

0

The algorithm works for compressible flows without modification. This is because  $\rho_c$  in Equation (1) can be an arbitary function of position, which was already stated - but the earlier sentence that the referee pointed out indeed implies something different. Gone through the manuscript making sure that this is clear.

**4. Streamline the figures**

Figures 2 and 3 convey nearly the same information. I suggest keeping the one that best illustrates the tracer-density evolution and moving the other to the supplement or removing it entirely.

Figures 2 and 3 are really complementary because by comparing the two figures the reader can see clearly how a tracer distribution converts to a tracer density field (something that is not normally plotted in publications and therefore unfamiliar to the readers). So I would argue that it's best to keep both figures close to each other in the main text.

------ RC2 ------

The study by P. Tackley presents a new algorithm for tracer advection for the marker-in-cell method. The algorithm corrects the displacement of tracers by solving for mass conservation of tracers. It is a neat, physics-based approach; however, I find that the choice of the potential function requires more discussion, and, if this algorithm is to be useful to the geodynamics community—or more broadly, to anyone using the marker-in-cell method, the writing can be improved and more tests would be useful. I detail these points below. Considering that the revision work I propose is considerable, I recommend a major revision of the manuscript.

1. Choice of the potential function for displacement vector in Eq. 3.

It is not clear why the author chose Eq. 4 (instead of Eq. 7) besides convenience to solve a linear Poisson equation. The justification is not there. I agree that it is more convenient to define dx=grad(phi)/rho\_t, but what is the physical meaning of this equation? What would be the physical units of such a displacement? Maybe the author has thought about this, but the displacement should not depend on tracer density, and only Eq. 7 is a physical choice. Neither Eq. 4 nor Eq. 9 is balanced in terms of units. Has the author tried solving the non-linear Poisson equation in Eq. 8?

The meaning of Equation (4) is clearer if written as

$$\rho_t \Delta \vec{x} = \nabla \phi$$

The left-hand side is mass flux, thus  $\phi$  is a mass flux potential. I have renamed it in the text and rewritten equation (4) in this form.

As the continuity equation is about div(mass\_flux) and this correction algorithm redistributes mass (to acheive the correct density), it is logical and physical to use a mass flux potential. The (SI) unit of displacement is metres and the units of mass flux above are kg/m², the physical meaning being kilograms flowing through a 1 m² area in one correction step. (In the continuity equation, mass flux is per second rather than per correction step). The units of  $\phi$  are kg/m and equations (4) and (9) are dimensionally consistent with this. I have added the units to the text.

The alternative definition of  $\phi$  used in equations (7) and (8) is a displacement potential with units of m2. I have changed the symbol to  $\phi$  so that it's a clearly different quantity.

The example code is nondimensional so there are no units.

L77, Eqs. 7-8: Why is the non-linear equation problematic in areas where rho\_t is zero? Not sure this is an issue because in the marker-in-cell method, zero tracer density in control volume is a violation, so this is avoided (which makes the models in Fig. 2 somewhat artificial).

Discretization of partial differential equations such as (8) generally leads to a matrix problem of the form:

[coefficients][unkowns] = [rhs]

where [coefficients] is a square matrix, [unknowns] is a column vector containing the quantity being solved for at each grid point (in this case  $\phi$ ) and [rhs] is a column vector of the know fixed right-hand-side. If too many coefficients are 0 then the matrix is singular and thus no solution is possible. That is the case here. The presence of rho\_t=0 in one or more cells results in a singular matrix and thus no solution.

Yes in principle the marker-in-cell method should have at least one marker in each cell, something that this method is intended to enforce. But after an advection step, it can be that there are cell(s) with no tracers and therefore this method would not work.

Minor comments regarding mathematical theory:

L46, Eq. 1: Need definition (equation) for tracer density in the main text. It was obtained from the code but was not obvious. Density = #nlocal\_mark \* nx\*ny\*nz / #ntracers

Equally, for the mean density, which is taken equal to 1 here.

Mean density = nmarkx\_cell\*nmarkz\_cell \* nx\*ny\*nz / #ntracers = 1

It was already stated that tracer density is mass of tracers per unit volume. Depending on the application, this could be nondimensional mass and density (as in the example code, this is why the mean value is 1.0) or dimensional mass and density (kg/m³), in which case the code line pasted by the referee would be different. I have added more explanation of this.

L47: tracer density is "defined as the mass (or number) of tracers per unit volume". This statement is correct only if it is assumed that all tracers have the same mass, and the density relates to the number of tracers. This needs to be clarified.

This is an "or" statement, it's not both at the same time.

People use the marker-in-cell approach in different ways. Sometimes markers do not have mass, in which case the "number density" is the relevant quantity, and this is what Pusok et al. (2016) plot, for example; actually they used number per cell rather than number per volume. Sometimes markers do have mass, in which case the usual "mass density" is the relevant quantity. In the latter case, markers can have different masses. The method works for all of these possibilities without changing the equations in the manuscript. The only difference would be that the units of  $\phi$  are different if using "number density" instead of "mass density", as I now point out. I have clarified these things.

Eq. 2 to Eq. 3: the discrete version is missing for v=dx/dt.

Most likely the reader can follow this step, but I now point this out just in case.

L55-57, Eq. 1: rho\_c is mean density of tracers in this study. To generalise, better to use 'initial' instead of 'correct' (i.e., rho\_0). Also for the purpose of this study, rho (without the t subscript) can be used for the tracer density.

In general, rho\_c can vary with position (see response to RC1 above) and can also change with time - it does not have to be the initial or the mean density. Therefore it is best to leave the subscripting as is. I have made this clearer in the text.

2. Generalisation and stopping criterion for nudging

L111: would the equations change for variable grid spacing?

No change. Equations (1)-(9) are physical so there is no change to them when the grid spacing varies, just as there is no change to the continuity, Stokes or energy equations.

Also, is the scheme generalisable for other boundary conditions, other than impermeable (L72)? Can the boundary conditions for Poisson be generalised from the velocity boundary conditions?

Yes. Given some equation(s) for the velocity boundary condition, equation (4) can be used to derive the relevant equation(s) for  $\phi$ . I now point this out.

The number of 'nudges' required seems arbitrary and it is not clear what is the computational cost of each nudge. I wonder if a stopping criterion can be derived? Example, norm(density error)<=tolerance, where tolerance = 1e-1 or 1e-2 from Fig. 4. Is that an overkill for using the algorithm?

A good idea, as a nudge might not be necessary every time step.

People don't normally complain about the initial tracer distribution in geodynamic models so a good tolerance to use seems to be the initial L1-norm of the tracer density error, which for the second test is about 3.5e-2 (see Figure 6). Thus, I have added tests (new Section 4.6) to see how frequently a correction needs to be made for the prescribed-cell-flow test (Section 4.2) and the new convection test (Section 4.5). Indeed, for this tolerance a correction is not needed every time step, saving CPU time.

**3. Geodynamic and performance tests**

The tests presented are highly simplified, and maybe artificial (Fig. 2). To demonstrate better the impact of the algorithm, I recommend adding another test showing a geodynamic problem—with sharp velocity interfaces or rotational flows near corners. It would be interesting to see, for example, the problem cited in L27 on eruption/intrusion, which was stated, but not explained, and how this algorithm deals with it.

Two more tests have been added (Section 4.3-4.4). Eruption/intrusion algorithms will be the topic of a future manuscript.

Then, there is the issue of the computational cost of this algorithm (with nudges). Simple Matlab scripts run fast, but the marker-in-cell method becomes expensive for 2-D and 3-D problems. What is the cost of this algorithm? How does it vary with grid size (resolution)?

L201-202: says the computational cost of solving Poisson is small, but need to demonstrate. For example, using a geodynamics test case, show time to solve for Stokes and time for tracer correction (Poisson\*number nudges).

**Now done (Sections 4.5-4.6).**

**4. Focus and style of writing**

The algorithm corrects the displacement of markers by solving for conservation of tracer mass. This does not transpire from the nudging and bunching described. The writing is too focused on the algorithm (which comes out almost as a trick to deal with marker dispersion/clustering), and not so much on the physics or why is it better than other methods. I think it is not doing a great job advertising why this could be a useful technique for geodynamics, and the informal style of writing does not help (e.g. usage of regular+, Euler+nudge+random without clear definition).

Other examples,

L40: the goal shouldn't be to nudge the tracers, but to correct their displacement in order to keep uniform density.

L46: the goal is to preserve the initial tracer distribution.

The Abstract should also be revised because I had the following questions:

L9: is the correction method physics-based? A sentence describing the method is lacking. Only results are described.

L14: what does the author refer to as non-conservation errors? How do they occur? They are introduced somewhat late, even in the abstract.

Excellent idea to start by stating that the mass conservation equation is used to derive the corrections, which are needed because of tracer advection not perfectly respecting the mass conservation equation. So I have now reworked the relevant introductory sections and all these points should be addressed now.

The details on the content of the Matlab routines in Section 3 (L112-118) and Fig 1. should be in an appendix. Instead, the author can outline a pseudo-algorithm such as

- Initialise tracer distribution, rho = rho 0
- Time loop:
- advect tracers with velocity field v,
- calculate rho and density error,
- correct tracer location by solving for displacement dx (Poisson equation).

I'd like to do what is normal and customary. Some published GMD papers have actual code (not pseudo-code) in the main text as well as details of how the code works, so this does seem acceptable. Examples:

Räss et al. (2022): https://doi.org/10.5194/gmd-15-5757-2022

Cheng et al. (2025): https://doi.org/10.5194/gmd-18-5311-2025

Therefore I leave it as it is.

Minor comments

L6-7, 17: tracers may track other fields besides composition and temperature throughout the domain. Their advantage is that they may perform better at advecting these fields compared to grid-based methods. L7: why does the problem occur? i.e., rotating fields in a box domain or at sharp interfaces. Addressed in the reworked introductory sections.

L8: Suggest to use 'clustering' or 'accumulation' instead of 'bunching'. I'm not finding a reason why one of these is better => leave as is.

L20: (e.g., \cite not \citep) Changed

L21: why all the major codes use marker-in-cell? It is better to justify advantages rather than state a common practice. Addressed in the reworked introductory sections.

L24: (e.g. 5-50) not necessary. The accuracy of the marker in cell technique relies on having a high density of tracers in each cell. I disagree, as someone who is new to the method might wonder whether "high density" means e.g. 10 or 100000.

L29: why the errors occur in the first place? L30: physics-based remedy is required Addressed in the reworked introductory sections.

L31: one previous solution is to create new tracers, but why is this not a good remedy? It's unphysical.

L44, L96: just because it is already implemented in StagYY, it is not a statement of validation or verification. StagYY should cite the work in the current manuscript to demonstrate the method, not the other way around. Already-published StagYY papers cannot cite this paper, so the point here is mainly to reassure the reader that some already-published papers benefit from this method. Anyway I reduced it to one statement.

L109: first time density bounds are given; this should occur earlier when density is defined. The magnitude of tracer density error is very application-dependent, so belongs where the results are presented.

L149: unclear statement. Tracers at the boundaries were nudged only a fraction of the calculated displacement to avoid crossing the boundaries. Not quite. Tracers that are initially calculated as crossing boundaries are instead moved only 70% of the calculated distance. Now stated like this.

Eqs. 11-12: could plot either the stream function or plot the velocity field to understand why the center particles might get dispersed. The stream function is a sine (0->pi) in x multiplied by the same in z so looks almost circular giving a simple "one cell circulation pattern". This seems too simple to be worth plotting particularly as I already doubled the number of figures.

Section 4.2 notation: need partial differentials (\partial). Equation (10) changed accordingly.

Eq. 12: mistake  $Vx = 1/Lz \sin() \cos()$  Fixed

L172: define Euler+nudge+random Defined

L174: calculated as in Eq. 12 (not above) Changed

Figures 2 and 3 do not have colorbars or state that blue points represent tracers. Figure 2: Colorbar not needed. Added that each blue dot is a tracer. Figure 3: Added colorbar

Fig 5: what is the computational cost or runtime between the cases? Add colorbars. See Sections 4.5-4.6. Colorbar added.

Fig 6: all cases should start with the same initial condition, and then additionally a case with completely random distribution. The missing case was added.

Matlab: Add in readme how to run the script. I.e. change ntest = 1; choice between 1-11. Added a README.txt

---

## Author Response (AR2)

**Minor points**

L38: to correct non-conservation errors; repeated "mass conservation". Changed

L52: to achieve an uniform tracer density Changed to "correct" rather than uniform (it's not uniform when compressible)

L53: sentence could be revised to be more concise or split into two sentences. Split.

L110: italic "and" Fixed

L121, 176, 213 (and throughout manuscript): revise mathematical notation for nx, ny, nz, S(x,z), vx, vz, etc. Fixed (I'm sure the copy editor will find anything that I missed)

Section 4.5, Table 1 caption: The problem being solved and for which these timings are reported is not explicitly stated (i.e., the thermal convection test in Section 4.4). Added

Fig 9 caption: Non-dimensional temperature and viscosity fields... Added

Notation in section 4.6: scientific  $2x10^{-2}$  versus exponential 2e-2. Changed to scientific.

Conclusion: the first sentence could be broken into smaller sentences. Done